# Learning Basketball Using Direct Instruction and Tactical Game Approach Methodologies

**DOI:** 10.3390/children8050342

**Published:** 2021-04-26

**Authors:** Sergio González-Espinosa, Javier García-Rubio, Sebastián Feu, Sergio J. Ibáñez

**Affiliations:** 1Facultad de Ciencias del Deporte, Universidad de Extremadura, 10003 Cáceres, Spain; jagaru@unex.es (J.G.-R.); sibanez@unex.es (S.J.I.); 2Facultad de Educación, Universidad de Extremadura, 06006 Badajoz, Spain; sfeu@unex.es

**Keywords:** teaching method, physical education, basketball, Tactical Game Approach

## Abstract

This study was to analyze and compare the learning acquired by the students in the sport of basketball in two different methodologies. The sample was composed of 40 students divided into two groups. The intervention programs had previously been validated. A descriptive analysis of the learning indicators, a t-test for independent samples to identify the differences between the methods, and a t-test for related samples to analyze the differences in each group were performed. There are differences between the performance profiles of students in the Direct Instruction in Basketball program and those in the Tactical Game in Basketball program in nine variables. Significant differences are found in the situations of dribbling, shooting, passing and movement, spacing, off-ball defense, and help and in the performance indicator for decision making, execution, and total, which are favorable to the Tactical Game in Basketball program. The students of the Direct Instruction in Basketball program only improved in three variables after the program, while the Tactical Game in Basketball students improved in thirteen variables. It is recommended that the teachers at the schools use the Tactical Game in Basketball methodology for their basketball teaching lessons, because student learning is better than in the Direct Instruction in Basketball program.

## 1. Introduction

Physical education (PE) teaching has evolved in the last few decades, and studying different teaching methodologies has helped in this evolution. Teaching methodologies can be divided into two main approaches: the teacher-centered approach (TCA) and the student-centered approach (SCA). TCA methodologies focus the teaching process on the teacher, while in SCA methods, the students are the protagonists of the learning process.

Direct instruction (DI) is a typical methodology of the TCA approach. In this method the teacher is the center of learning development, and controls all the activities that are carried out, making planning very important for this methodology. This involves the detailed planning of the task presentation and structure, the total time to be devoted to the task, the different play areas to be used, and the necessary equipment [1]. The DI methodology looks for high levels of opportunities to respond (OTR), and the key is the presentation and structuring of the tasks. Eight dimensions have been indicated for improving the presentation and structuring of the tasks: making instructions explicit, emphasizing the usefulness of the content being presented, structuring new content, signaling students’ attention, summarizing and repeating information, checking for understanding, creating a productive climate for learning, and presenting accountability measures [2].

The teaching sessions using this methodology follow a progression according to difficulty. The initial tasks are isolated to be subsequently incorporated into play situations [3]. The evolution towards more complex play situations occurs when the teacher considers that the students have sufficiently mastered the technique [4]. The initial sessions use task structures like individual practice in self-space, individual practice in repetitive drills, teacher-led practice, or low-organizational games [1]. The means used to initiate training in this method are simple or complex application exercises [5,6]. As well as planning and structuring the tasks, the communication between the teacher and the students is vital to develop the PE contents. The corrective feedback that the teacher provides for the students in the DI method is descriptive, centered on the differences with the model to be followed. The feedback can be given during the task itself or between the tasks which make up the session [4].

The game centered approach (GCA) “advocates learners playing the game as the central organizational feature of a lesson” [7]. This approach has been called a “wave to the future” [8]. One of the most commonly used methodologies is the Tactical Game Approach (TGA). The TGA method [9] was devised as a modification of Teaching Games for Understanding (TGfU) [10]. The TGA methodology simplifies the six stages for the acquisition of tactical awareness proposed by TGfU (play, perception of play, tactical awareness, decision making, technical execution, and performance). TGA simplifies this process into three phases: form of play (real or exaggerated), tactical awareness (what should I do), and skill execution (how should I do it) [9]. The structures of the tasks are based on forms of play where a tactical problem has to be solved. This type of task obtains a multitude of possible responses to the problem set and ways of suitably practising their solutions [11]. The tasks have to have their own presentation and structure, but they should all be centered on the solution to a tactical problem. The teacher is responsible for presenting a tactical problem that has to be developed using a series of tasks. The sequence should begin with a game form that helps the students to discover what problem has to be solved. Then tasks are designed to work on the tactical and technical needs observed in the first task, which can be repeated to ascertain whether the skill has improved. Subsequently modified games are introduced, which exaggerate the tactical problem to be resolved and follow the evolution of the tactical problem proposed at the beginning of the session [9]. Finally, there is a situation of a real game, where the development of the tactical problem worked on in the session is observed. The use of the feedback provided by the teacher is vital in this methodology for the students’ progress. The teacher uses interrogative feedback so that the students autonomously develop their decision making and create their own tactical awareness [9]. The feedback can be classified into five categories depending on the set objective: time, space, risk, “what”, and “how” [1].

Many studies have attempted to identify the benefits of using the different teaching methodologies. The most commonly studied variables have been related to declarative or procedural knowledge and the students’ behavior in the game [12,13,14,15]. Furthermore, several studies have worked on affective variables according to the teaching method [13,16].

In spite of the large volume of research on teaching methodologies, the literature suggests continuing with studies to analyze the different methodologies and their contribution to the teaching–learning process [17]. Therefore, the objective of the present study was to analyze and compare the learning acquired by the students in the sport of basketball after the implementation of a program based on the DI methodology or a program based on the TGA methodology, analyzing game performance before and after the implementation of the intervention program in each of the methodologies. The intervention programs in the classroom were validated for being applied in the school. The assessment instrument analyzed the learning in a real game situation, according to the basic components of the sport: decision making, technical execution and effectiveness. The evolution of the students’ learning was also analyzed for each intervention program.

## 2. Materials and Methods

### 2.1. Teaching Experiment

Teaching experiments refer to studies centered on the curriculum or the educational environment in which the researchers work on the contents, the curriculum and the teaching–learning process in the school context [18]. The design of the teaching experiment in this study was of 12 sessions for teaching basketball to two groups, one group using the direct instruction and the other the tactical game methodology. The assessment of the students was based on decision making, technical execution, and effectiveness of the different basketball actions. A pre-test and a post-test were performed to assess the improvement of the students after the teaching sessions. The teacher selected to develop the sessions was a member of the research group developing this study. This researcher was acquainted with the teacher’s intervention for both methodologies. Apart from the theoretical training for developing the intervention, the researcher did a pilot test beforehand, consisting of a replica of the Tactical Game in Basketball (TGB) and Direct Instruction in Basketball (DIB) programs and an identical procedure to a previous investigation [19]. The TGB is a specific program for teaching basketball based on the Tactical Games Approach.

### 2.2. Participants

Forty students between 11 and 12 years of age participated in the study. The students were divided into two groups, group A (20 students, 11 boys and nine girls) working with the DI methodology and group B (20 students, 10 boys and 10 girls) working with the TGA method. The groups were formed by each of the classes. The students belonged to an infants and primary school in the south west of Spain. None of the students had prior experience of formal basketball sports training. The pretest results show that there are no significant differences between the TGA teaching methodology class and the DIB teaching methodology. Therefore, we can affirm that they are homogeneous groups and without previous experience in the sport of basketball. The teacher who imparted both programs was a qualified primary school physical education teacher. All the parents/tutors of the students who participated in the study gave their informed consent. The study complies with the ethical principles stated in the Declaration of Helsinki and was approved by the Ethics Committee of the University.

### 2.3. Design and Procedure

The present study was quasi-experimental with a randomized group design [20]. Each group was given a different program for teaching basketball; in one group, the DIB unit was imparted using DI methodology, and in the other group, the TGB unit was based on TGA (independent variables) [19]. Basketball performance (dependent variable) was measured before and after the application of the teaching program in all students with an assessment test in a 3 × 3 game situation before beginning (pre-test) and at the end (post-test). A total of 12 sessions were given, ten sessions of 50 min each to develop each unit and two assessment sessions, pre-test and post-test, of 50 min each. All sessions were developed within the formal school hours of the Physical Education subject. The sessions of each of the methodologies are independent. The teacher only performs the session with one of the groups and in another lesson time with the other group.

In the Spanish educational system, children go to school for calendar years, but academic courses take place over two calendar years, from September to June. Therefore, during the school year students, may be 11 or 12 years old. Each experimental group orclass conducted their sessions during the class timetable set by the school management. The class groups never coincided in the school playground.

### 2.4. Teaching Programs

#### 2.4.1. Direct Instruction

All the tasks were validated beforehand for suitability and preparation for the teaching methodology by a panel of experts. To be part of the panel of experts, four of the following criteria had to be met: (1) be a doctor; (2) be or have been a university professor; (3) possess the highest federation qualification in a collective sport; (4) have 10 years of experience as a university teacher; (5) have 10 years of experience as a team sports coach; and (6) have publications on teaching–learning and/or sports training methodologies. The unit based on DI lasted ten sessions divided into two parts: an initial set of five lessons for individual technical execution and another five lessons for the execution of individual and group movements in team play. During the first five sessions, they worked on the layup from the right, left, and center. Several tasks were also performed on individual technique, and the different types of dribbling and individual on-ball defense. In the final five lessons, analytical tasks were performed on spacing as a function of the game situation (see Table 1).

The teacher was the model in all the tasks because of his excellent mastery of the subject, having been a professional player. The students were placed in rows and repeated one after the other the model explained by the teacher. The teacher was the main manager of the lesson, both with regard to organization and to the contents to be developed in the tasks. The teacher’s feedback was descriptive, defining success or failure according to how close the student had come to the model demonstrated.

Students were grouped together freely and voluntarily at the beginning of the class to carry out the tasks. During the Physical Education class, in addition to the development of specific contents, we also worked on attitudinal contents, values, such as equality or companionship. For this reason, during the development of the session, the Physical Education teacher proposes rotations and changes of partners between tasks. Only during the tests were the groupings of students stable.

#### 2.4.2. Tactical Game Approach

As in the previous method, all the proposed tasks were validated by the same group of 17 experts. This methodology also lasted ten teaching sessions. As with the DI methodology, it was divided into two parts: the five initial sessions were focused on individual play and the final five sessions on team play. In the first lesson, they worked on dribbling skills and basket shots with offensive advantage to favor a greater number of actions ending in a basket shot. Sessions two, three, and four were focused on the contents of basket shots and 1 vs. 1 situations. In lesson five, the content of individual defense was added to the previous tasks. In lessons six to ten, the contents were centered on team situations: offensive numerical superiority and situations of equal number (see Table 1).

The presentation of the tasks was simple, it was only necessary to explain the organization of the task and the objective to be attained. The students were the focus of the learning and had to decide how to reach the objective set by the teacher. During the tasks, the teacher analyzed the students’ mistakes to be able to try to correct them later using questions. The teacher used interrogative feedback with the aim of getting the students to modify their erroneous actions according to the final objective to be achieved and to work on the individual technique needed to be successful.

#### 2.4.3. Validity

The DIB and TGB teaching programs were validated [21] by a panel of 17 expert referees who judged the suitability and preparation of each intervention program. All the tasks, except two, of those that comprised both intervention programs surpassed the values of validity required according to Aiken’s V (Aiken 1985) (V > 0.70) for the study. The two tasks that did not fulfil the requisites were modified. The reliability of the programs measured with Cronbach’s α [20] was excellent α = 0.927 in the DIB program and 0.945 for the TGB.

### 2.5. Instruments

#### 2.5.1. Basketball Learning and Performance Assessment Instrument

The instrument to measure learning and performance in basketball (BALPAI by its Spanish initials) was designed to provide teachers and coaches with a valid and reliable instrument for assessing performance in play for training stages [22]. This tool assesses a total of 11 play actions, which are dependent variables in the study. A total of seven offensive play actions (dribble, pass, shoot, reception of the ball (reception), pass and move, spacing, and offensive rebound (off reb)), and another four actions for defensive play (defensive rebound (def reb), on-ball defense (on-ball def), off-ball defense (off-ball def), and help and recover (help)). For each play action, the BALPAI assesses the performance indicator (PI) for decision making (DM), execution (EX), effectiveness (EF), and total performance. These components determine the success or failure of an action and thus performance. These play actions come from the taxonomy of contents proposed by Ibáñez [23]. The instrument establishes three levels of adequacy for each of the components of the play actions: inadequate, neutral, or adequate. In the assessment, each of the play action components is awarded one point (inadequate), two points (neutral), or three points (adequate) to be able to calculate the average for each play action as a function of the number of actions performed. The assessment makes it possible to obtain the PI of DM, EX, and EF, and the mean is used to calculate the total PI.
Total PI = (DM + EX + EF)/3

#### 2.5.2. Validity and Reliability

A total of 13 expert referees validated the BALPAI. Thus, all the experts had to fulfill at least four of the following six criteria: (i) have a Ph.D. in Sports Sciences; (ii) be or have been a university lecturer; (iii) have the highest federative qualification in a team sport; (iv) have 10 years’ experience as a university lecturer; (v) have 10 years’ experience as a team sport coach in any category, and (vi) have published articles on the topic of team sports. They analyzed each of the variables in the instrument for their adequacy and preparation. All the values obtained an Aiken’s V > 0.75, confirming that the instrument is valid for measuring performance in the game.

Three observers intervened in the study of the reliability of the instrument, and they were unaware of the number of cases that had to be distributed by category [24]. The Free-Marginal Multirater Kappa (Multirater Kfree) was used, and the values of reliability obtained for the instrument were α > 0.81, which according to Ladis and Koch [25] and Altman [26] demonstrate the almost perfect reliability of the instrument.

### 2.6. Data Collection

Before and after the administration of the program, the students performed an assessment test to measure their performance in basketball. The play situation selected for the tests was 3 vs. 3 with one basket. This competitive format was the best suited to the characteristics of the sample [12,27,28]. The teams were formed randomly, but with the same number of players of each sex per team. The competition was a round robin, thus eliminating a possible contaminating variable like the level of the opponent. The students’ performance was calculated with the BALPAI that analyzes different dimensions of each play action: decision making, execution, and effectiveness.

### 2.7. Data Analysis

First tests for normality, equality of variance, and randomness were performed to select the model for testing the hypothesis [20]. The descriptive analysis of the mean and standard deviation was used to characterize the sample. A t-test for independent samples was carried out to determine the differences between the pre-test and post-test in both learning programs [29]. A *t*-test was also performed for related measures to calculate the improvement in each group between the pre-test and post-test. In both cases, the effect size was calculated [30] with Cohen’s d to complement the inferential analyzes. All the data analyses were performed with the SPSS 21.0 (SPSS Inc., Chicago IL, USA) statistical program. Statistical significance was set at *p* < 0.05.

## 3. Results

Figure 1 shows the performance profile of the students of both groups according to the program they were going to follow after the initial assessment. Figure 1 is complemented with the table of results (Table 2), which compares the results of the pre-test.

It is evident that before the intervention programs, the performance profiles were similar. The students from both groups performed better in play actions like shooting, passes, and reception. Moreover, in the performance variables, DM got a higher score than the other variables. No significant differences were recorded in the play actions except in spacing, where the students who were going to follow the TGB method performed better than the DIB group before the intervention.

Figure 2 shows the performance profile of the DIB and TGB groups after the interventions in each of the programs. Figure 2 is also complemented with the results of the t-test for independent samples in the post-test (Table 3).

There are differences between the performance profiles of the students in the DIB program and those in the TGB program (Figure 2). The greatest differences are found in the dribble, shooting, pass and move, spacing, off-ball defense, and help. The actions of shoot and pass are the ones that obtained the highest values using both methodologies. In contrast, the def reb is where the TGB group obtained its lowest score, and def reb and pass and move are the actions where the DIB group recorded their lowest score. All the PI show higher values in the TGB program, and the DM PI is the variable with the highest values in both programs.

In the results of the post-test (Table 3), it can be seen that after the intervention, the students from the TGB unit showed significant differences to those of the DIB unit in the dribble, shooting, reception, pass and move, spacing, on-ball defense and off-ball defense. The total PI for the TGB unit showed higher PI in DM, EX, and total variables, revealing a better performance in all the play variables except the defensive rebound.

Table 4 presents the differences between the pre-test and post-test of the students who followed the DIB program.

The results of the t-test for related samples of the students in the DIB unit show a significant improvement in on-ball and off-ball defense. Furthermore, these students significantly improved their effectiveness in the play actions. In spite of the fact that only three variables obtained a level of statistical significance (*p* < 0.05), the rest of the variables, except the defensive rebound, improved their mean with regard to the pre-test.

Table 5 shows the differences between the pre-test and the post-test in the group that followed the TGB program.

The comparative results of the pre-test and post-test in the students trained with the TGB unit show how they improved in almost all the skills. There are significant differences after the implementation of the program. In spite of the improvement, the help and def reb variables did not improve in the TGB group.

## 4. Discussion

The purpose of this study was to analyze and compare the learning acquired in basketball by the students after the implementation of an intervention program with a different teaching methodology in each group. The results of the study show that the students who received the TGB program obtained better results than those who followed the DIB unit, with more improvements between the pre-test and the post-test in almost all the variables. The students from the DIB unit only improved in three of the variables measured in the study.

The students taught with the TGB method obtained better results in total performance that those who followed the DIB unit. These findings are similar to those of other studies that compared TCA methodologies with SCA methodologies [12,31,32]. One of the most important aspects for the development of each unit is the communication between teacher and students. Although descriptive intervention is recommended for students when they are facing new contents [33] the results of this investigation do not support this premise. The DI method used descriptive communication and obtained worse results than the TGA methodology. The TGA methodology uses a reflective intervention [9] with which the student progressively acquires autonomy in the game [34]. The difference between these two types of intervention, descriptive and reflective, may be due to the fact that the sport analyzed is one of collaboration–opposition and not an individual sport. In basketball, it has been shown that better DM helps in recognizing a tactical situation more quickly and effectively. This quicker and more effective DM favors enhanced performance in the game [35,36]. These results correspond to those found in our study. The students who obtained better DM results also recorded a better total performance in the game, with a significant difference with respect to the students from the DIB unit. Therefore, it can be confirmed that autonomous learning favors DM in students.

Several investigations have not found differences in technical execution using different approaches [15,37,38], and the assessment tests used by the researchers have been targeted as the reason why. These assessments have been decontextualized tests in which technical execution is recorded in a play action. This type of assessment separates the phases of perception and decision making in the students under study. The separation from real play and the exclusion of perception and decision making in the assessment of execution signifies that these tests do not have ecological validity [17]. In the present study, technical execution was assessed in a situation of real play, in competition. The instrument used therefore possesses great ecological validity, as it does not remove the execution from the real context of play nor from the phases of perception and decision making. The differences shown between the DI and the TGA approaches can be considered more valid and reliable than in previous studies, as the results show the differences in a real way. The findings present a significant difference in the PI of EX in favor of the students from the TGB unit in comparison to those from the DIB unit.

The students from the DIB unit only improved in three variables, while the students from the TGB unit showed more improvements. This may be due to the fact that in the classes using the DIB method, the useful practice time for the students is very short [39]. The tasks are organized around the repetition of a technique, and the number of repetitions is low, as the teacher has to continuously supervise each student’s performance [39]. This type of organization meant that the students in the DIB program had little or no decision making to do during the tasks set [40]. On the other hand, the students from the DIB group did significantly improve their on-ball and off-ball defense. Defense can be classified as a more closed play situation than attack. It is thus understandable that being an action where decision making is less important, the use of prescriptive feedback was more effective than in more open situations.

The advantage of the TGA methodology is not only to be found in basketball performance. In the literature, a transfer of tactical awareness has been found in categories of similar games, collaboration–opposition sports [41], and also in the movement patterns of technical skills that are often common to many physical domains [42,43]. The development of more mature movement patterns is favored by the TGA methodology, as the tactical and technical skills are approached in a context of learning centered on play [37,44].

The research carried out by Gamero et al. (In press) [45] shows that schoolchildren who follow defined sports teaching programs at school with different approaches, either the Direct Instruction Approach or the Tactical Games Approach, improve their learning compared to those who follow undefined programs. These results are similar to those of this research. Students improve with both programs, but the improvement is greater when learning takes place under a Tactical Games Approach.

## 5. Conclusions

This study suggests the use of TGA methodology for learning basketball in comparison to DI methodology. Furthermore, after the application of the TGA methodology, there is an improvement in the decision making and technical execution variables. In the variables specific to basketball, this study obtained better results with the TGA methodology in the following variables: dribble, shooting, reception, pass and move, spacing, and on-ball and off-ball defense. There were no differences in the pass and off-rebound variables between both groups, but the TGA group did significantly improve these variables in the pre-post comparison. One of the limitations of this quasi-experimental study is the extrapolation of the results in a general way to the whole educational system, as the sample of participants is of a specific age group. Nevertheless, it provides PE teachers with scientific evidence on the suitability of one methodological approach over another.

This line of research should continue to be developed in other groups of students of different ages, as well as in the context of sport, at initial training levels.

## Figures and Tables

**Figure 1 children-08-00342-f001:**
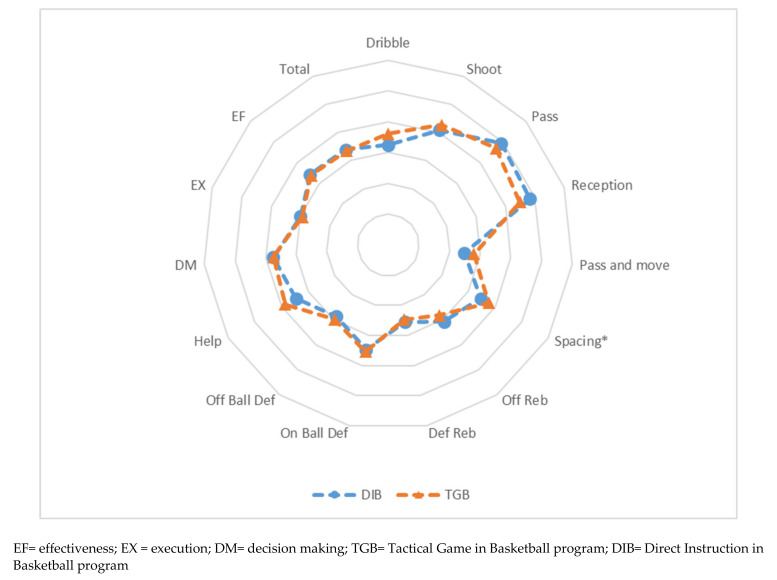
Characteristics of the students after the pre-test. * *p* < 0.05.

**Figure 2 children-08-00342-f002:**
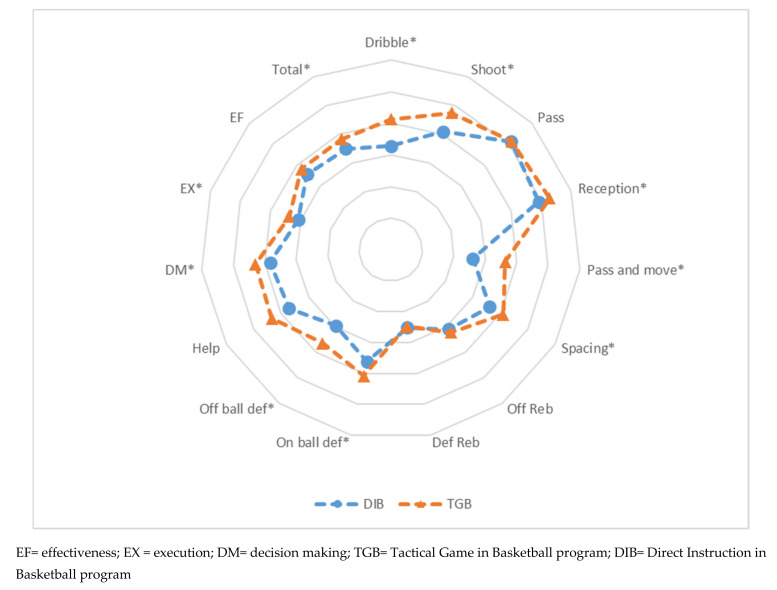
Characteristics of the students after the post-test. * *p* < 0.05.

**Table 1 children-08-00342-t001:** Unit plans for the two instructional approaches.

Lesson	DIB	TGB
1	- Technical execution of the different dribbling techniques. - Representation and imitation of the basket shot with approaching steps from the right, left, and center.	- Dribble tag. Tag using dribbling. - Discovery of the approach to the basket and shot in situations of offensive advantage in 1 vs. 1.
2–4	- Repetition of the basket shot after approaching steps from the right, left and center. - Basket shot with approaching steps after overcoming an obstacle.	- Moving to the basket and shooting in situations of offensive advantage in 1 vs. 1. - Decision making to shoot or dribble in 1 vs. 1.
5	- Repetition of the basket shot after approaching steps from the right, left, and center. - Basket shot with approaching steps after overcoming an obstacle. - Individual on-ball defense technique.	- Moving to the basket and shooting in situations of offensive advantage in 1 vs. 1. - Decision making to shoot or dribble in 1 vs. 1. - Discovery of placement and position in on-ball defense.
6	- Basket shot with approaching steps after overcoming an obstacle. - Jump shot from the front. - Offensive movements for offensive numerical superiority 2 vs. 1. - Offensive movements for 2 vs. 2.	- Moving to the basket and shooting in situations of offensive advantage in 1 vs. 1. - Decision making to shoot or dribble in 1 vs. 1. - Decision making with offensive numerical superiority 2 vs. 1. - Initiation to team play: 2 vs. 2.
7	- Technical execution of the different types of passes. - Offensive movements for offensive numerical superiority 2 vs. 1. - Offensive movements for 2 vs. 2 play.	- Tag with different passes. - Decision making with offensive numerical superiority 2 vs. 1. - Team play 2 vs. 2.
8	- Individual technique in team defense. - Offensive movements for numerical superiority 3 vs. 2. - Offensive movements for 3 vs. 3 play.	- Individual defense in situations of team play. - Decision making in offensive numerical superiority 3 vs. 2. - Initiation to team play: 3 vs. 3.
9	- Individual technique in on-ball defense. - Offensive movements for 2 vs. 2 play. - Offensive movements for 3 vs. 2 numerical superiority. - Offensive movements for 3 vs. 3 play.	- Placement and position in on-ball defense. - Team play 2 vs. 2. - Decision making with offensive numerical superiority 3 vs. 2. - Team play 3 vs. 3.
10	- Basket shot with approaching steps after overcoming an obstacle. - Offensive movements for 2 vs. 2 play. - Offensive movements for 2 vs. 1 numerical superiority. - Offensive movements for 3 vs. 3 play.	- Decision making to shoot or dribble in 1 vs. 1. - Team play 2 vs. 2. - Decision making to shoot or dribble in 1 vs. 1. - Team play 3 vs. 3.

DIB = Direct Instruction in Basketball; TGB = Tactical Game in Basketball.

**Table 2 children-08-00342-t002:** Differences in performance between students in the DIB and TGB units in the pre-test.

		TGB		DIB							
		*M*	*SD*	*M*	*SD*	*T*	*p*		*gl* ^1^	*gl* ^2^	*d*
Play actions	Dribble	1.804	0.337	1.624	0.283	1.824	0.076		1	38	0.578
	Shoot	2.132	0.231	2.039	0.382	0.932	0.357		1	38	0.295
	Pass	2.347	0.284	2.467	0.249	1.427	0.162		1	38	0.449
	Reception	2.245	0.336	2.414	0.324	1.622	0.113		1	38	0.512
	Pass and move	1.396	0.377	1.241	0.227	1.578	0.123		1	38	0.498
	Spacing	1.883	0.152	1.734	0.135	3.279	0.002	*	1	38	0.038
	Off Reb	1.406	0.190	1.537	0.314	1.592	0.120		1	38	0.505
	Def Reb	1.239	0.158	1.275	0.175	0.688	0.496		1	38	0.216
	On-ball def	1.769	0.247	1.735	0.304	0.382	0.705		1	38	0.123
	Off-ball def	1.488	0.199	1.433	0.185	0.898	0.375		1	38	0.286
	Help	1.934	0.499	1.730	0.597	1.175	0.247		1	38	0.371
PI	DM	1.872	0.197	1.878	0.177	0.113	0.911		1	38	0.032
	EX	1.466	0.184	1.496	0.176	0.518	0.607		1	38	0.167
	EF	1.686	0.174	1.713	0.212	0.448	0.657		1	38	0.139
	Total	1.675	0.181	1.696	0.184	0.364	0.718		1	38	0.115

TGB = Tactical Game in Basketball program; DIB = Direct Instruction in Basketball program; * *p* < 0.05; *gl*
^1^ = number of groups less one; *gl*
^2^ = number of participants less one; EF = effectiveness; EX = execution; DM = decision making.

**Table 3 children-08-00342-t003:** Differences in performance between the students in the DIB and TGB units in the post-test.

		TGB		DIB							
		*M*	*SD*	*M*	*SD*	*t*	*p*		*gl* ^1^	*gl* ^2^	*d*
Play actions	Dribble	2.064	0.306	1.645	0.245	4.790	0.000	*	1	38	1.512
	Shoot	2.372	0.186	2.048	0.390	3.342	0.002	*	1	38	1.60
	Pass	2.569	0.240	2.562	0.214	0.093	0.926		1	38	0.031
	Reception	2.637	0.204	2.460	0.304	2.158	0.037	*	1	38	0.684
	Pass and move	1.820	0.416	1.305	0.251	4.745	0.000	*	1	38	1.499
	Spacing	2.053	0.242	1.803	0.156	3.879	0.000	*	1	38	1.228
	Off Reb	1.620	0.337	1.552	0.312	0.662	0.512		1	38	0.209
	Def Reb	1.241	0.120	1.261	0.162	−0.426	0.673		1	38	0.140
	On-ball def	2.039	0.197	1.811	0.260	3.131	0.003	*	1	38	0.988
	Off-ball def	1.829	0.236	1.483	0.145	5.568	0.000	*	1	38	1.767
	Help	2.170	0.454	1.858	0.658	1.750	0.088		1	38	0.552
PI	DM	2.153	0.179	1.917	0.181	4.147	0.000	*	1	38	1.311
	EX	1.691	0.220	1.539	0.177	2.398	0.021	*	1	38	0.761
	EF	1.903	0.206	1.781	0.179	1.984	0.055		1	38	0.632
	Total	1.916	0.199	1.746	0.176	2.857	0.007	*	1	38	0.905

TGB = Tactical Game in Basketball program; DIB = Direct Instruction in Basketball program; * *p* < 0.05; *gl*
^1^ = number of groups less one; *gl*
^2^ = number of participants less one; EF = effectiveness; EX = execution; DM = decision making.

**Table 4 children-08-00342-t004:** *T*-test for related samples in the DIB program.

		Pre-Test		Post-Test							
		*M*	*SD*	*M*	*SD*	*t*	*p*		*gl* ^1^	*gl* ^2^	*d*
Play actions	Dribble	1.624	0.283	1.645	0.245	−0.387	0.703		1	19	0.068
	Shoot	2.039	0.382	2.048	0.390	−0.344	0.734		1	19	0.025
	Pass	2.467	0.249	2.562	0.214	−1.731	0.100		1	19	0.366
	Reception	2.414	0.324	2.460	0.304	−0.828	0.418		1	19	0.137
	Pass and move	1.241	0.227	1.305	0.251	−1.319	0.203		1	19	0.270
	Spacing	1.734	0.135	1.803	0.156	−1.910	0.071		1	19	0.491
	Off Reb	1.537	0.314	1.552	0.312	−0.465	0.647		1	19	0.046
	Def Reb	1.275	0.175	1.261	0.162	1.052	0.306		1	19	0.080
	On-ball def	1.735	0.304	1.811	0.260	−2.304	0.033	*	1	19	0.239
	Off-ball def	1.433	0.185	1.483	0.145	−2.478	0.023	*	1	19	0.260
	Help	1.730	0.597	1.858	0.658	−1.002	0.329		1	19	0.206
PI	DM	1.878	0.177	1.917	0.181	−1.673	0.111		1	19	0.211
	EX	1.496	0.176	1.539	0.177	−1.709	0.104		1	19	0.236
	EF	1.713	0.212	1.781	0.179	−2.497	0.022	*	1	19	0.311
	Total	1.696	0.184	1.746	0.176	−2.097	0.050		1	19	0.262

TGB = Tactical Game in Basketball program; DIB = Direct Instruction in Basketball program; * *p* < 0.05; *gl*
^1^ = number of groups less one; *gl*
^2^ = number of participants less one; EF = effectiveness; EX = execution; DM = decision making.

**Table 5 children-08-00342-t005:** *T*-test for related samples in the TGB program.

		Pre-Test		Post-Test							
		*M*	*SD*	*M*	*SD*	*t*	*p*		*gl* ^1^	*gl* ^2^	*d*
Play actions	Dribble	1.804	0.337	2.064	0.306	−3.151	0.005	*	1	19	0.742
	Shoot	2.132	0.231	2.372	0.186	−4.133	0.001	*	1	19	0.997
	Pass	2.347	0.284	2.569	0.240	−4.210	0.000	*	1	19	0.751
	Reception	2.245	0.336	2.637	0.204	−7.479	0.000	*	1	19	1.120
	Pass and move	1.396	0.377	1.820	0.416	−5.439	0.000	*	1	19	1.080
	Spacing	1.883	0.152	2.053	0.242	−3.785	0.001	*	1	19	1.073
	Off Reb	1.406	0.190	1.620	0.337	−3.091	0.006	*	1	19	1.079
	Def Reb	1.239	0.158	1.241	0.120	−0.075	0.941		1	19	0.015
	On-ball def	1.769	0.247	2.039	0.197	−4.521	0.000	*	1	19	1.050
	Off-ball def	1.488	0.199	1.829	0.236	−9.112	0.000	*	1	19	1.643
	Help	1.934	0.499	2.170	0.454	−2.036	0.056		1	19	0.454
PI	DM	1.872	0.197	2.153	0.179	−9.558	0.000	*	1	19	1.372
	EX	1.466	0.184	1.691	0.220	−7.893	0.000	*	1	19	1.169
	EF	1.686	0.174	1.903	0.206	−8.655	0.000	*	1	19	1.199
	Total	1.675	0.181	1.916	0.199	−9.556	0.000	*	1	19	1.275

TGB = Tactical Game in Basketball program; DIB = Direct Instruction in Basketball program; * *p* < 0.05; *gl*
^1^ = number of groups less one; *gl*
^2^ = number of participants less one; EF = effectiveness; EX = execution; DM = decision making.

## Data Availability

The authors have the data available for the journal if it is required.

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
