# Peer review of "Learning Basketball Using Direct Instruction and Tactical Game Approach Methodologies"

_children, 2021, doi:10.3390/children8050342_

Round 1

Reviewer 1 Report

11. correct the words "two different methodologies" 15, 16, 19. Define the acronyms DIB, TGB and TGA 16. Make clear that 'significant differences' are between before and after TGB 18. Reword this sentence as is suggests that DIB has 3 improvements but TGB has none. 19. Make clear why TGA is recommended above DIB Consider slight reworking of the abstract to make clear the distinction between TGA and TGB and present the results concisely. 24. Explain exactly what evolution has occurred. 103. Consider adding a sentence addressing the issues surrounding insider researcher and researcher/practitioner bias on. 116. (comment, no change required) Excellent consideration of ethical concerns gives a lot of credibility to this study 131 and 146. Expand upon what is meant by "experts'. 164. Is this the same as panel of experts as mentioned in 131 and 146? 192. Again, make clear if this panel of 13 experts is related to the 17 experts in line 164. 324. Consider adding a sentence address the limitations of the study (perhaps including researcher practitioner bias and the size of the research cohort). This is a well written, interesting and credible academic study. The attention to detail and statistical analysis gives confidence to the reader. Slight amendments to the Abstract will ensure potential readers are fully informed regarding the quality of this study. The suggestions made, which are minor additions, are not critical but would add further polish to this excellent research paper. Thank you for the opportunity to review your paper and I wish you every success in your future research.

Author Response

Thank you very much for your corrections to improve the article. We have answered all your suggestions in the attached letter. 

Reviewer 2 Report

L1:the word ” twodifferents” needs to insert space

L110: The development of children is relatively different. In addition to supplementing whether the 40 children are in the same class or the same grade, please also describe in detail the background information of the children participating in the study, such as gender, month age... ..

L111: Please clearly describe how the 40 students were divided into two groups in this study and the background information after grouping

L113: Please clearly state whether the course attribute in this study is a formal school course or a free elective course after class or a club course

L128:In the 2.4.1 (DI) and (TGA) teaching plans, there are many 1v.s1 or 2v.s2 group activities in different ways. Please explain how these children are divided into groups during the course

L138:Please add that in the 50-minute course, how does the teacher carry out two different teaching plans for children aged 11-12 at the same time?

L324: In children’s life experience, play is a very important basic element in the learning process. Therefore, the (TGB) teaching method will have its own learning incentives. Therefore, when this research uses (TGB) and (GIB) two teaching methods When conducting research, it is necessary to put forward effective suggestions for children to learn basketball under these two teaching methods in order to show the specific value of this research. Conclusions and suggestions should be provided for the advantages and disadvantages of the two teaching methods on the basic skills of basketball, so as to help the quality of the implementation of children's basketball courses.

Author Response

(The authors gave the same response as above.)
